# Spatiotemporal Evolution and Regional Differences in the Production-Living-Ecological Space of the Urban Agglomeration in the Middle Reaches of the Yangtze River

**DOI:** 10.3390/ijerph182312497

**Published:** 2021-11-27

**Authors:** Yanqiong Zhao, Jinhua Cheng, Yongguang Zhu, Yanpu Zhao

**Affiliations:** 1School of Economics and Management, China University of Geosciences, Wuhan 430074, China; zhuyongguang@cug.edu.cn; 2School of Earth Resources, China University of Geosciences, Wuhan 430074, China; ypzhao1993@163.com

**Keywords:** production-living-ecological space, Theil index, ESDA, regional differences

## Abstract

The urban agglomeration in the middle reaches of the Yangtze River, which is the second largest urban agglomeration in China, represents a typical land space range of ecological vulnerability in China. Large differences occur in economic development mode between resource- and non-resource-based cities in this basin area. Accurate identification of the evolution and regional differences in the production-living-ecological space (PLES) is very important in order to elucidate the development and utilization of land space in the region. At present, relevant research has largely focused on the classification and determination of PLES temporal and spatial patterns. Temporal and spatial pattern research has mainly considered a single scale of administrative division, whereas fewer studies have analyzed the temporal and spatial patterns and regional differences in the PLES in ecologically fragile natural watersheds. Therefore, based on PLES classification, the regional differences in the PLES between two types of cities in the basin are measured via the Theil index and exploratory spatial data analysis (ESDA). First, the ecological space (ES) of these two types of cities in the urban agglomeration in the middle reaches of the Yangtze River is compressed by the production space (PS) and living space (LS), in which the ES of resource-based cities is compressed for a longer period, and the phenomenon involving PS compression by the LS and ES mainly occurs in non-resource-based cities within the urban agglomeration in the middle reaches of the Yangtze River. Second, the PLES of these two types of cities exhibits the characteristics of spatial aggregation, and high- and low-density areas of the PLES remain relatively stable. Third, the regional differences in the PLES of the urban agglomeration in the middle reaches of the Yangtze River mainly originate from intraregional differences. The PLES of these two types of cities in the urban agglomeration in the middle reaches of the Yangtze River is more sensitive to changes in economic development than to those in the population distribution.

## 1. Introduction

Since the reform and opening up, rapid economic growth in China has led to tense human–Earth relations, and the phenomenon of production-living-ecological space (PLES) competition has increasingly occurred [1,2]. Disorderly development of national land can restrict regional sustainable development, cause a series of social problems likely leading to a decline in regional development competitiveness and affect ecological, investment and living environments [3,4,5,6]. The report of the 18th Communist Party of China (CPC) National Congress introduced the land space optimization goal entailing the promotion of intensive and efficient production spaces (PSs), appropriate living spaces (LSs) and beautiful ecological spaces (ESs), which further improved the Chinese land space optimization theoretical system based on the PLES [7]. The opinions on ecological civilization construction acceleration issued by the State Council in 2015 further emphasized the PLES, which provided a theoretical basis for the study of the evolution of land spatial patterns and functions [8]. The report of the 19th CPC National Congress in 2017 proposed to complete the demarcation of three control lines, including the ecological protection red line, permanent basic farmland boundary and urban development boundary, which aimed to coordinate the spatial pattern relationship between PSs, LSs and ESs and promote sustainable development of the urban economy and environment [9]. The effective mitigation of PLES contradictions and realization of coordinated PLES development have become the focus of attention in land development in China [10,11].

The PLES theory is based on the perspective of the element structure function in systematics [12]. Elements and structures comprise the basis to realize the basic functions of the land space [13,14]. Analysis of the evolution of the spatial form and structure constitutes a means to explore the change process of spatial functions [15,16,17,18]. According to the functional attributes of spaces, the PLES theory divides the land space into the PS, LS and ES [19,20,21,22]. The LS is the spatial carrier of the daily life activities of people, the PS is the specific functional area established by people engaged in production activities, and the ES is the regional space providing ecological products and services [23,24].

Moreover, the evolution of national land and regional differences are the result of man–land interrelations [25,26]. The change process of structures entails the change in space over time (dynamic process) and the projection of time in space, namely, the traces left by spaces in historical development and change processes [27,28]. The change process occurs in a relatively static state. Spatial evolution is the result of the variation in space forms and structures [29].

At present, research on the PLES focuses on three directions: type division, spatiotemporal pattern determination and spatial optimization of the PLES. Spatial classification of the PLES comprises the basis for research on PLES spatiotemporal patterns and spatial optimization, which has been reported in the existing literature [30,31]. Starting from the primary functions of various land use types, scholars have compiled a catalogue of the land use types of the PLES [32,33]. However, scholars have considered the cross functions of certain types of land use and have introduced the LS-ES, ES-PS and ES-PS types to ensure parallelism with the PLES, which has resulted in a very complex PLES classification process [34,35].

Scholars have examined PLES patterns at different scales. At the national scale, the reclassification assignment method has been applied in ArcGIS software to explore the spatial distribution pattern of the PLES. Research has indicated that PSs are mainly distributed in the main urban agglomeration and grain-producing areas on the eastern side of the Hu Huanyong line, LSs largely occur in the main cities and urban agglomerations in China, and ESs are mostly located on the western side of the Hu Huanyong line [8,36]. At the provincial scale, grid analysis, spatial autocorrelation analysis, comprehensive PLES evaluation, trigonometric analysis of a single dynamic degree and spatial analysis have been employed to study the spatial distribution pattern and evolution characteristics of a given province [37,38]. Research has indicated that there are obvious regional differences in the spatial pattern and evolution of the PLES within a province [37,39]. Research on Hubei Province has demonstrated that the PS and LS have obviously expanded, and the ES has obviously contracted [40]. Research on Henan Province has revealed that the LS and ES have tended to remain stable, while the PS has greatly changed [29]. At the urban agglomeration scale, the PLES function measurement model has been applied in empirical analysis. PSs and LSs are concentrated in eastern coastal areas and exhibit an expanding trend, while ESs are concentrated in western mountainous and hilly areas and exhibit a decreasing trend overall [10]. At the prefecture-level city scale, the grid kernel density method has been employed to analyze the PLES pattern of a megacity (Wuhan) [38]. At the county scale, the spatial Lorentz curve and structural Gini coefficient [9], spatial transfer matrix [41], neighborhood analysis [42], spatial autocorrelation [43] and hotspot analysis [44] have been applied in empirical analysis of the PLES spatial pattern in different counties. In addition to the above administrative spatial scale, studies have exceeded administrative boundaries to explore the spatial pattern of PLES functions in river basins (Bailong River Basin) [23], deltas (the core area of the Yangtze River Delta) [45], mountainous areas (Taihang Mountain area) [46] and ecotourism areas [10].

Considering that the interactions between the PS, LS and ES are intensifying, with the transformation from functional zoning of a single element to territorial zoning covering an entire region, many scholars have examined the territorial space from the PLES perspective, carried out research on ecotourism areas [47] and coastal zones [6], and greatly contributed to territorial space development and land regulation. In terms of research methods, functional zones have been defined according to quantitative methods, such as the comparative advantage index [48] and double-constraint clustering [49], or by combining quantitative and qualitative methods, such as advantage function identification, double clustering and expert qualitative adjustment [50]. The thousand-layer cake model [51], classification and evaluation index system of the PLES [52] and ecological evaluation model [53] have been adopted to formulate PLES optimization strategies at different spatial scales.

The researches on the spatial-temporal pattern and regional differences of the PLES of natural watersheds provide the basis for the land space control and natural watershed governance. Previous studies have important reference values for this paper, but most of them just studied the classification and temporal and spatial pattern of PLES. Furthermore, the research on the temporal and spatial pattern focused on the single scale of administrative division, such as national, provincial and urban scale. Few studies have analyzed the temporal and spatial pattern and regional differences of PLES in ecologically fragile natural watersheds. In this study, we aimed to reveal the temporal and spatial pattern and regional differences of the PLES of urban agglomerations in the middle reaches of the Yangtze River. Another aim was to make proposals for eco-environmental protection and high-quality development in the Yangtze River Basin.

## 2. Materials and Methods

### 2.1. Study Area

The urban agglomeration in the middle reaches of the Yangtze River is a giant urban agglomeration mainly comprising the Wuhan Urban Circle, Changsha Zhuzhou Xiangtan Urban Agglomeration and Poyang Lake Urban Agglomeration. Rapid urbanization of the urban agglomeration in the middle reaches of the Yangtze River imposes an increasingly prominent coercive effect on the ecosystem, which seriously threatens the sustainable development process of this urban agglomeration. Therefore, scientific measurement of the temporal and spatial evolution characteristics of the PLES of the above urban agglomeration in the middle reaches of the Yangtze River could be highly important for the formulation of ecosystem health protection policies for this urban agglomeration.

At the first National Symposium on resource-based cities in 2018, relevant experts pointed out that according to existing relevant research, comparative analysis of resource- and non-resource-based cities and determination of the similarities and differences between these two types of cities are urgently required, which could better guide regional sustainable development. Based on PLES classification, this paper examines the evolution of the PLES of these two types of cities (resource- and non-resource-based cities) (Figure 1.) in the urban agglomeration in the middle reaches of the Yangtze River and adopts the Theil index to measure the regional differences in the PLES between these two types of cities, which can clarify the similarities and differences between these two types of cities in the development and utilization of national land and resources.

### 2.2. Data Source

The data in this paper include Chinese land use/land cover remote sensing monitoring data (land use/cover change (LUCC), 1-km accuracy) pertaining to 1995, 2000, 2005, 2010 and 2015. ArcGIS software is employed to extract PLES land use data of county-level cities in the urban agglomeration in the middle reaches of the Yangtze River, which are considered to analyze the spatial and temporal patterns and regional differences in the PLES among county-level cities in the Yangtze River Basin. The selected land use data are retrieved from the Resource and Environmental Science Data Center of the Chinese Academy of Sciences (http://www.resdc.cn/ accessed on 20 September 2021). Gridded Chinese population distribution and GDP data with a spatial resolution of 1 km for 1995, 2000, 2005, 2010 and 2015 are acquired from the Data Center of Resources and Environmental Science, Chinese Academy of Sciences (http://www.resdc.cn/ accessed on 20 September 2021), extracted with ArcGIS software. Both types of data include the effects of the land use type, night light brightness and residential area density, which ensures that the Theil index can more accurately reflect the evolution of PLES patterns and regional differences with the GDP and population as weights. Choosing the GDP and population distribution as weights considers the impacts of economic development and population distribution differences, respectively, on the PLES (Table 1).

### 2.3. Classification of the Production-Living-Ecological Space (PLES)

By comparing current research results on PLES classification, considering the concept of the PLES, this paper comprehensively obtains lessons in terms of the classification methods of the PLES from previous research [31,50] (no space types are considered other than the PS, ES and LS in the primary classification system of the PLES). The current LUCC classification system in China has been integrated and reclassified [48]. This classification system is summarized in Table 2.

### 2.4. Theil Index

The Theil index was proposed by Theil in 1967. This index mainly examines the matching degree between the population and income and further assesses the degree of regional differentiation. Subsequently, this index was widely adopted in relevant research on geography, economics and sociology. Decomposition of the Theil index can reveal the change direction and amplitude of interregional differences (provinces, cities, districts and counties) and intraregional differences. Moreover, this approach can calculate the contribution rate of intraregional and interregional differences to the total differences. The Theil index is adopted to calculate the regional differences in the PLES in the urban agglomeration in the middle reaches of the Yangtze River, and the calculation equation of the Theil index of the PLES is expressed as:(1)T=∑i=1nSiSlog(Sisqiq)

In Equation (1), T denotes the Theil index of the PLES of the urban agglomeration in the middle reaches of the Yangtze River, S denotes the total land use area of the PLES of the urban agglomeration in the middle reaches of the Yangtze River, Si denotes the land use area of the PLES of the resource- and non-resource-based cities in the urban agglomeration in the middle reaches of the Yangtze River, q denotes the GDP or population of the urban agglomeration in the middle reaches of the Yangtze River, and $q_{i}$ denotes the GDP or population of the resource- or non-resource-based cities. When q denotes the GDP, the calculated Theil index considers the impact of economic development on the regional differences in the PLES. When q refers to the population, the impact of the population spatial distribution on the regional differences in the PLES of the urban agglomeration in the middle reaches of the Yangtze River is considered.

The Theil index achieves a good decomposition effect. When the sample is divided into multiple groups, the Theil index can be divided into intergroup and intragroup gaps, which can measure the difference between regions. According to the basic concept of Theil index decomposition and drawing on the results of previous studies, the cities in the urban agglomeration in the middle reaches of the Yangtze River are divided into resource- and non-resource-based cities. The area and regional development interregional and intraregional differences are calculated based on the Theil index of the PLES. The total Theil index of the urban agglomeration in the middle reaches of the Yangtze River is divided into the intraregional and interregional Theil indices.
(2)Tr=∑i=1nSiSrlog(SiSrqiqr)
(3)Tg=∑i=1nSiSglog(SiSgqiqg)
(4)$$T_{w}=\frac{S_{r}}{S}\;T_{r}+\frac{S_{g}}{S}T_{g}$$
(5)Tb=SrSlog(SrSqrq)+SgSlog(SgSqgq)
(6)$$T=T_{w}+T_{b}$$

In Equations (2) and (3), $T_{r}\;\mathrm{and}\;T_{g}$ denote the Theil indices of the PLES of the resource- and non-resource-based cities, $S_{i}$ denotes the area of a certain type of PLES in the ith city, $q_{i}$ denotes the GDP or population of the ith city, Sr denotes the area of a certain type of PLES of the resource-based cities in the urban agglomeration in the middle reaches of the Yangtze River, and $S_{g}$ denotes the area of a certain type of PLES of the non-resource-based cities in the urban agglomeration in the middle reaches of the Yangtze River. In Equations (4) and (5), $T_{w}$ denotes the Theil index of the two types of urban areas, $T_{b}$ denotes the Theil index of the region between the two types of urban areas, and T denotes the total Theil index of the PLES of these two types of cities.

Based on Theil index decomposition, various contribution rates to the Theil index were measured to reflect the impact of the spatial differences between the resource- and non-resource-based cities on the differences in the PLES of the urban agglomeration in the middle reaches of the Yangtze River. According to Equation (6), dividing the left and right terms by T yields the following:(7)$$1=\frac{T_{w}}{T}+\frac{T_{b}}{T}=\frac{\frac{S_{r}}{S}T_{r}}{T}+\frac{\frac{S_{g}}{S}T_{g}}{T}+\frac{T_{b}}{T}$$
(8)$$CT_{w}=\frac{T_{w}}{T}$$
(9)$$CT_{b}=\frac{T_{b}}{T}$$
(10)$$CT_{r}=\frac{\frac{S_{r}}{S}T_{r}}{T}$$
(11)$$CT_{g}=\frac{\frac{S_{g}}{S}T_{g}}{T}$$

In Equations (8)–(11), $CT_{w}$ and $CT_{b}$ denote the contribution rates of the interregional and intra-regional differences, respectively, to the regional differences in the urban agglomeration in the middle reaches of the Yangtze River, and $CT_{r}$ and $CT_{g}$ denote the contribution rates of the resource- and non-resource-based cities, respectively, to the regional differences in the urban agglomeration in the middle reaches of the Yangtze River.

### 2.5. Exploratory Spatial Data Analysis (ESDA) Method

Exploratory spatial data analysis (ESDA) encompasses a collection of spatial data analysis methods. ESDA adopts the spatial relevance as the core, obtains the spatial agglomeration aspects and anomalies of objects through visual description of the spatial dependence and spatial heterogeneity in data, and explains the spatial relationship between regions through the definition of the spatial weight matrix. Moreover, ESDA reveals the spatial action mechanism and evolution rules between objects. ESDA mainly relies on two methods. One method is global spatial autocorrelation, typically employed to evaluate the distribution characteristics of objects or attribute values in the whole space and determine whether adjacent areas exhibit aggregation characteristics, which is generally detected by global Moran’s I index. The second method is local spatial autocorrelation, which is employed to analyze the distribution characteristics of subsystems and measure the spatial differences between a specific region and adjacent regions. Local spatial autocorrelation is generally measured with local Moran’s I index.

(1) Global spatial autocorrelation, which can be expressed as follows:(12)$$Moran^{\prime }s\;I=\frac{\sum_{i=1}^{n}\sum_{j=1}^{n}W_{ij}\left(X_{i}-\bar{X}\right)\left(X_{j}-\bar{X}\right)}{S^{2}\sum_{i=1}^{n}\sum_{j=1}^{n}W_{ij}}$$

In Equation (12), n is the total number of all counties and cities in the urban agglomeration in the middle reaches of the Yangtze River, i≠j, and $W_{ij}$ is either 0 or 1. In this paper, all counties and cities in the urban agglomeration in the middle reaches of the Yangtze River are selected, $\bar{X}=\frac{1}{n}\overset{n}{\underset{i=1}{\sum }}X_{i}$ is the average of the observed values, $W_{ij}$ is the spatial weight matrix, $X_{i}\;\mathrm{and}\;X_{j}$ are the observed values of i and j, respectively, and $S^{2}=\frac{1}{n-1}\overset{n}{\underset{i=1}{\sum }}{\left(X_{i}-\bar{X}\right)}^{2}$ is the variance in the observed values.

The value of global Moran’s I generally occurs in [–1,1]. I<0 indicates that there exists a negative correlation in the urban agglomeration in the middle reaches of the Yangtze River. When the I value is close to −1, cities with different attributes in the urban agglomeration in the middle reaches of the Yangtze River are clustered (low and high values or high and low values occur adjacent). For I>0, this indicates a positive correlation. When the value approaches 1, cities with similar attributes in the urban agglomeration in the middle reaches of the Yangtze River are clustered (low values occur adjacent to low values or high values occur adjacent to high values). I=0 indicates no spatial autocorrelation.

(2) Local spatial autocorrelation is calculated as follows:(13)$$Local\;Moran^{\prime }s\;I=\frac{\left(X_{i}-X\right)}{S^{2}}\underset{j\neq 1}{\sum }W_{ij}\left(X_{j}-X\right)$$

For *Local Moran’s I* > 0, this suggests that the high value of a given regional city in the Yangtze River urban agglomeration is surrounded by high values (high-high), or a low value is surrounded by low values (low-low). For *Local Moran’s I* < 0, this indicates that the high value of a regional city in the Yangtze River urban agglomeration is surrounded by low values (high-low), i.e., a hotspot area occurs, or a low value is surrounded by high values (low-high), i.e., a cold spot area occurs.

## 3. Experimental Results and Analysis

### 3.1. Evolution Trend of the PLES

According to the PLES classification system (Table 2), after reclassification of the land use type data pertaining to the urban agglomeration in the middle reaches of the Yangtze River in ArcGIS 10.4 software, the grid data are transformed into vector data, the PLES of the cities in the middle reaches of the Yangtze River is extracted, and the PLES of the above two types of cities is determined and mapped at different times.

Figure 2 shows that from 1995 to 2015, the PLES of the resource and non-resource-based cities in the urban agglomeration in the middle reaches of the Yangtze River changed dramatically. There were both differences and similarities in the evolution trend of the PLES of these two types of cities. The differences were as follows: from 1995 to 2005, the PS of the non-resource-based cities first increased and then decreased (inflection point, 2000), while that of the resource-based cities exhibited the opposite tendency. From 2005 to 2015, the ES of the resource-based cities first increased and then decreased (the inflection point occurred in 2010), while that of the non-resource-based cities exhibited the opposite behaviour. The similarities were as follows: from 2005 to 2010, the PS indicated an upward trend, and from 2000 to 2015, the LS continued to increase. Combined with the change trend of the PLES of these two types of cities, it could be determined that from 1995 to 2005, the synchronous increase in the PS and LS of the resource-based cities compressed the ES. From 2005 to 2010, the PS and LS of the non-resource-based cities increased, thereby compressing the ES, while from 2010 to 2015, the PS and LS of the resource-based cities simultaneously increased, thus compressing the ES. The LS and ES of the non-resource-based cities increased simultaneously, thus compressing the PS. In summary, the ES of these two types of cities was compressed by the LS and PS, and the occurrence period for the resource-based cities was longer. The PS was compressed by the LS and ES, and this phenomenon occurred for a longer time in the non-resource-based cities. The former did not facilitate ecological and environmental protection of the urban agglomeration in the middle reaches of the Yangtze River, whereas the latter did not promote overall development of industry and agriculture in the middle reaches of the Yangtze River.

### 3.2. Analysis of the Spatial Aggregation of the PLES

#### 3.2.1. Global Moran’s I Index Analysis

In ArcGIS software, the global Moran I value of the PLES of all provinces and cities in the middle reaches of the Yangtze River was calculated, and the inverse distance method was adopted for spatial conceptualization. The inverse distance method adopted the Euclidian distance, and row standardization was used. Table 3 indicates that the global Moran I value of the PLES of all areas in the middle reaches of the Yangtze River from 1995 to 2015 was positive. Moreover, in each year, the significance test was passed at the different statistical levels. The PLES of the spatial area in the middle reaches of the Yangtze River exhibited obvious positive spatial aggregation characteristics from 1995 to 2015. In addition, from the perspective of time, the PLES of the urban areas in the middle reaches of the Yangtze River attained different changes and fluctuations at the different times during the measurement period. The spatial agglomeration effect of county-level cities near the middle reaches of the Yangtze River fluctuated.

#### 3.2.2. Local Moran’s I Index Analysis

The global Moran index revealed that the spatial area of county-level cities near the urban agglomeration in the middle reaches of the Yangtze River experienced a spatial agglomeration effect, but it did not reflect the specific structure of spatial agglomeration. This paper further applied the local Moran index to measure the spatial heterogeneity among the county-level cities in the urban agglomeration in the middle reaches of the Yangtze River. The measurement results are shown in Figure 3 and Figure 4. It can be observed that the high-density area (high-high category) remained stable, and the low-density area (low-low category) of the resource-based cities remained constant as a whole. The high-density area (high-high category) of the non-resource-based cities also remained stable.

We measured the degree of functional coordination spatially by local spatial auto-correlation analysis. The local spatial autocorrelation results of resource-based cities in the middle reaches of the Yangtze River show: high-high aggregation and low-low aggregation characteristics indicate that the functional coordination degree of a single county and surrounding counties is high. The functional coordination degree of the territorial space is in a relatively stable state (Figure 3 and Figure 4).

### 3.3. Analysis of the Regional Differences between the Three Types of Spaces of the Urban Agglomeration in the Middle Reaches of the Yangtze River

#### 3.3.1. Theil Index of the Two Types of Cities in the Urban Agglomeration in the Middle Reaches of the Yangtze River

Using the Theil index and its decomposition formula, we analyze the grid data of urban agglomeration in the middle reaches of the Yangtze River from 1995 to 2015, obtain the Theil index with GDP and population as the weight of the urban agglomeration in the middle reaches of the Yangtze River, and calculate the regional differences between the two types of cities. Based on the GDP and population distribution as the weight (Table 4), the Theil index value of the two types of cities with the GDP as the weight is higher than that with the population distribution as the weight. This indicates that in the urban agglomeration in the middle reaches of the Yangtze River, the spatial area difference between these two types of cities is much more notably influenced by the difference in the urban economic development level than by the difference in the population spatial distribution. From the perspective of both the resource- and non-resource-based cities, the Theil index value of the PLES of the non-resource-based cities is higher than that of the resource-based cities (Table 4). Regardless of whether the GDP or population is employed as the weight, the regional difference in the PLES of the non-resource-based cities is more notably affected by the GDP and population factors than that in the PLES of the resource-based cities, resulting in the above phenomenon. The reason may be that the economic development mode of the resource-based cities is singular, while the economic development mode of the non-resource-based cities is more diversified. The difference between the supply and demand levels of the PLES can exert a notable impact on the non-resource-based cities.

According to the change trend, with the GDP as the weight (Figure 5a,b), the change trend of the PLES of these two types of cities mostly indicates an upward–downward–upward trend pattern. Therefore, the PLES of these two types of cities can be divided in two stages. From 1995 to 2005, the PS difference between the resource-based cities first increased and then decreased, the PS difference between the non-resource-based cities increased, and the LS difference between the resource-based cities continued to increase. During the same period, the LS difference between the non-resource-based cities increased first and then decreased, and the ES difference between the resource-based cities continued to expand. Moreover, the ES difference between the non-resource-based cities exhibited a trend of first decreasing and then increasing. From 2005 to 2015, the differences in the PS and LS between these two types of cities first decreased and then increased, which was similar to the change trend of the ES differences between the resource-based cities, while the ES differences between the non-resource-based cities continued to decline. With the population as the weight (Figure 5c,d), from 1995 to 2005, the PS difference between the resource-based cities first decreased and then increased, the LS difference continued to decrease, and the ES difference indicated a trend of first increasing and then decreasing. Similar change trends of the PS, LS and ES differences between the non-resource-based cities were observed. From 2005 to 2015, the interregional difference in the PLES between these two types of cities expanded.

#### 3.3.2. Interregional and Intraregional Theil Indices of the PLES

In terms of the numerical value, the interregional differences across the urban agglomeration in the middle reaches of the Yangtze River are very small with either the GDP or population distribution as the weight, namely, smaller than 0.2, and most differences do not exceed 0.01 (Table 5), while the intraregional Theil index value is much higher than the interregional Theil index value. This suggests that regional differences are the main reason for the PLES variations in the urban agglomeration in the middle reaches of the Yangtze River.

A notable fluctuation is observed. Since the overall difference in the interregional Theil index of all cities mainly originates from the difference in the intraregional Theil index between these two types of cities, this paper further observes the change trend of the Theil index across the region (Figure 6). With the GDP as the weight (Figure 6a), from 1995 to 2005, the PS and LS exhibited an inverted V-shaped trend, and the regional differences first increased and then declined. In contrast, during the same period, the ES indicated a positive V-shaped trend, and the regional differences first declined and then increased. From 2005 to 2010, the regional differences in the PS and ES decreased (the inflection point occurred in 2000), while the regional differences in the LS increased. There was no significant change in the regional differences between these three types of spaces from 2010 to 2015. With the population as the weight (Figure 6b), from 1995 to 2005, the regional differences in the PLES exhibited an upward–downward–continuous upward trend pattern, with an inverted S-shaped distribution as a whole. From 1995 to 2005, the regional differences in the PLES first increased and then decreased (the inflection point was 2000). From 2005 to 2015, the regional differences in the PLES continued to increase.

#### 3.3.3. Contribution Rate of the Regional Differences between the Two Types of Cities

Numerically, considering the two weights, the contribution rate of the non-resource-based cities to the regional differences in the PS, LS and ES is higher than that of the resource-based cities, namely, CTr-PS > CTg-PS, CTr-LS > CTg-LS, CTr-ES > CTg-ES (Table 6). This indicates that the differences in the PLES of the non-resource-based cities exert a major impact on the regional differences in the PLES of the urban agglomeration in the middle reaches of the Yangtze River.

Based on the change trend, with the GDP as the weight, the contribution rates of the regional differences in the PLES of the resource-based cities exhibit an upward–downward–upward trend pattern. The contribution rate of the regional differences in the PS and ES of the non-resource-based cities indicate a downward–upward–downward trend pattern, while the contribution rate of the regional differences in the LS first decreases and then increases (Figure 7a,b). With the population as the weight, the contribution rate of the regional differences in the PLES of the two types of cities exhibits the opposite trend. In particular, the contribution rate of the regional differences in a certain type of PLES of the resource-based cities demonstrates a downward–upward–downward trend pattern, and the regional difference contribution rate of a certain type of PLES of the non-resource-based cities indicates an upward–downward–upward trend pattern (Figure 7c,d). Overall, the contribution rate of the regional differences in the PLES of the resource-based cities decreases, while the contribution rate of the regional differences in the PLES of the non-resource-based cities increases.

## 4. Discussion

In this study, we can conclude that in the urban agglomeration in the middle reaches of the Yangtze River, from 1995 to 2005, the synchronous increase of the PS and LS in resource-based cities squeezed the ES. From 2005 to 2010, the PS and LS of non-resource-based cities increased, squeezing the ES. From 2010 to 2015, the PS and LS of resource-based cities increased simultaneously, squeezing the ES. The LS and ES of the non-resource-based cities increased simultaneously, thus compressing the PS. The PLES of the two types of cities have the characteristics of spatial aggregation, and the density area is relatively stable. The regional differences in the PLES mainly originate from intraregional differences. Economic development can better disturb the changes of PLES.

Cui et al., by analyzing the evolution characteristics of PLES pattern in Hubei Province, concluded that from 2009 to 2015, the PS and LS in Hubei Province expanded significantly, and the ES shrank significantly [38]. The expansion of PS is mainly concentrated in the Wuhan urban circle, which is consistent with the conclusion that the PS of resource-based cities expanded and the ES was compressed from 2005 to 2015. Jin et al., through research on the evolution and function measurement of PLES of urban agglomeration in Fujian Delta, concluded that the ES of urban agglomeration in Fujian delta showed the overall change characteristics of obvious reduction rather than expansion from 2000 to 2015 [33]. The main reason was the rapid development of the social economy and the continuous expansion of urban land. This conclusion is consistent with the fact that economic development can more disturb the spatial change of PLES in this study. Yang et al., by studying the land use transformation and eco-environmental effects in the Yangtze River Delta, concluded that the production land decreased and the living land increased in the region from 1990 to 2010 [12]. In this study, the PS expansion in the middle reaches of the Yangtze River from 1990 to 2010. The reason may be that the development of agricultural production in the middle reaches of the Yangtze River is more developed than that in the Yangtze River Delta.

Combined with the above analysis, the following policy suggestions are formulated to provide a relevant basis for land and space control, ecological environment protection and sustainable development of the urban agglomeration in the middle reaches of the Yangtze River.

Against the strategic background of joint protection and the absence of large-scale development, we should strive to ensure prioritization of the ES, improve the occupation rate of the ES of these two types of cities, increase the ES through industrial migration or upgrading and population control, and reduce the frequency of human social and economic activities in the ES to alleviate the notable contradiction between human activities and the ecological vulnerability of the ES.

Economic growth exerts a notable impact on the differences between the three types of spaces in the urban agglomeration in the middle reaches of the Yangtze River. In the future, in regard to control of the three types of spaces of these two types of cities, we should pay more attention to the control of the three types of spaces of those cities with faster economic growth between these two types of cities.

Corresponding policies should be differentiated and formulated. According to the results of local Moran index, we can determine the density areas of various PLES in the two types of cities. Regarding PS and LS high-density areas, the improvement and transformation of the utilization of these two types of spaces should be strengthened. Regarding PD and LS low-density areas, long-term space utilization planning should be achieved to avoid space utilization waste. In terms of ES high-density areas, we should continue to maintain the stability of the ES across the region, and regarding ES low-density areas, we should improve the ES across the region.

## 5. Conclusions

Via processing of spatiotemporal data related to the PLES of the resource- and non-resource-based cities in the urban agglomeration in the middle reaches of the Yangtze River from 1995 to 2015, this paper examines the spatial evolution and spatial aggregation characteristics of these two types of cities and dynamically assesses the regional differences in the PLES between these two types of cities under the influence of economic development and population distribution with the Theil index. The main conclusions are as follows:

(1) From 1995 to 2015, the PLES of resource-based cities and non-resource-based cities in the middle reaches of the Yangtze River changed sharply. In different periods of this process, the following phenomena exist in both types of cities. The ES of was compressed by the PS and LS, and the ES of the resource-based cities was compressed for a longer period. That was not conducive to ecological environment protection of the urban agglomeration in the middle reaches of the Yangtze River. The LS and ES mainly compressed the PS. This phenomenon occurs for a longer time in non-resource-based cities, which did not facilitate economic production enhancement.

(2) From 1995 to 2015, the spatial aggregation characteristics of the PLES of the county-level cities in the urban agglomeration in the middle reaches of the Yangtze River were obvious, exhibiting positive spatial autocorrelation characteristics. From 1995 to 2015, the high-density areas (high-high category) and low-density areas (low-low category) of various PLES in resource-based cities and non-resource-based cities remained stable.

(3) The regional differences in the PLES of the urban agglomeration in the middle reaches of the Yangtze River mainly originated from the differences between urban areas. The PLES of the two types of cities in the urban agglomeration in the middle reaches of the Yangtze River was more sensitive to changes in economic development. The regional differences in the PLES of the non-resource-based cities are relatively large, while those in the PLES of the resource-based cities are relatively small. The possible main reason is that the economic development modes of the non-resource-based cities are diverse, while the development modes of the resource-based cities are relatively singular. The regional differences in the PS, LS and ES of the resource-based cities exert a major impact on the PLES of the urban agglomeration in the middle reaches of the Yangtze River.

In the study, Theil index and ESDA are used to explore the regional differences of PLES between the two types of cities under the influence of economic development and population distribution. However, the pattern, structure, form and relationship of the micro scale of PLES have not been studied in depth. In addition, the specific reasons leading to the change of PLES have not been explored in depth. For the above deficiencies, further exploration and research are needed.

## Figures and Tables

**Figure 1 ijerph-18-12497-f001:**
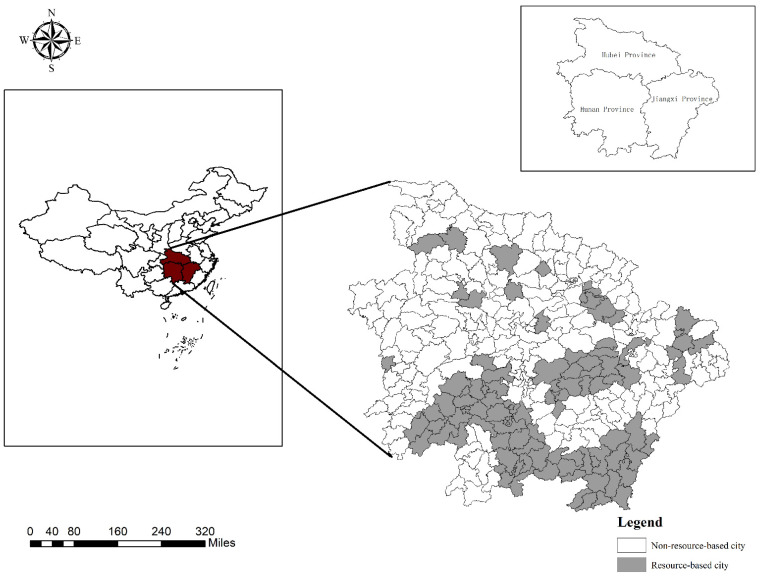
Spatial distribution of urban agglomeration in the middle reaches of the Yangtze River.

**Figure 2 ijerph-18-12497-f002:**
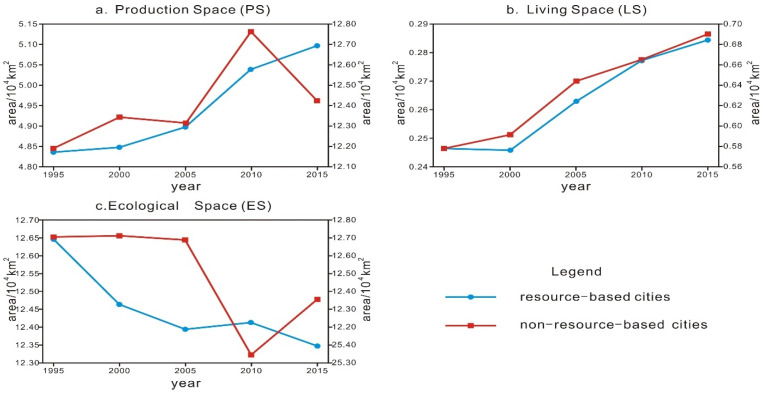
Change trend of the PLES of the two types of cities in the middle reaches of the Yangtze River. Note: The left longitudinal axis indicates the resource-based cities, and the right longitudinal axis indicates the non-resource-based cities.

**Figure 3 ijerph-18-12497-f003:**
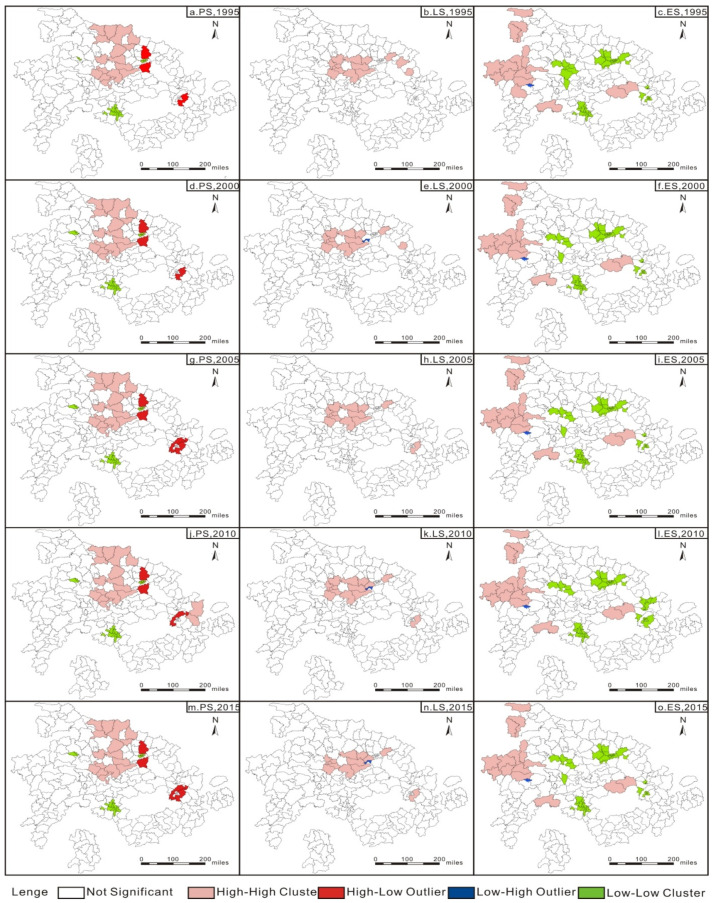
Local spatial autocorrelation results of the PLES of the resource-based cities, 1995–2015.

**Figure 4 ijerph-18-12497-f004:**
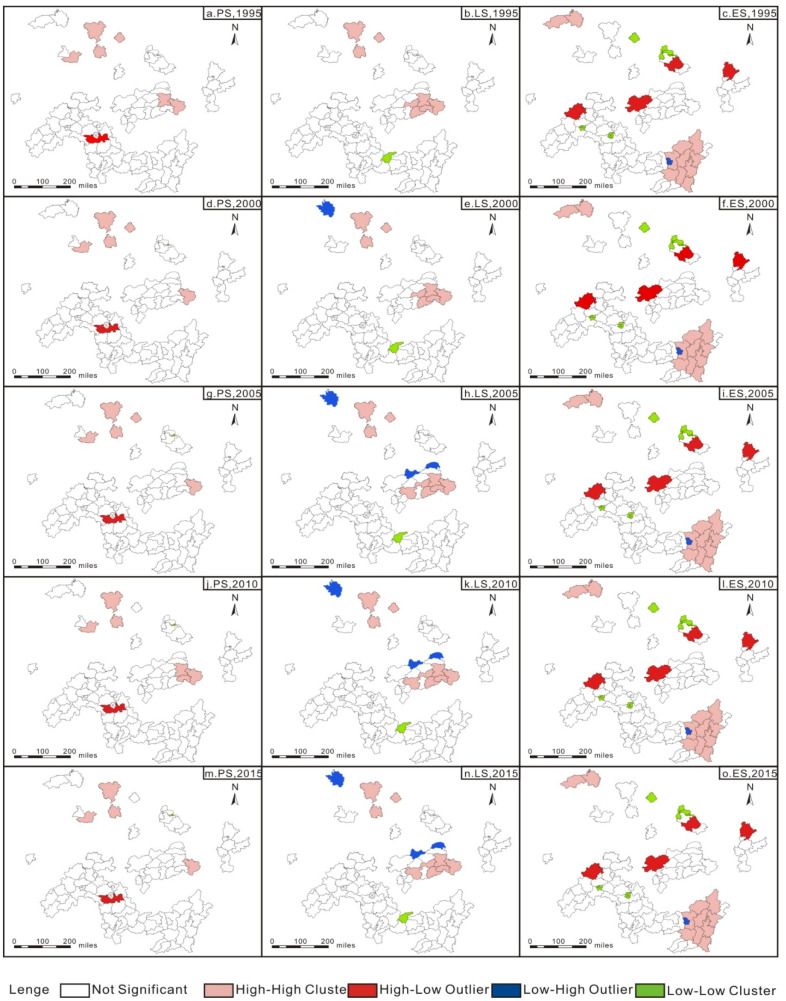
Local spatial autocorrelation results of the PLES of the non-resource-based cities, 1995–2015.

**Figure 5 ijerph-18-12497-f005:**
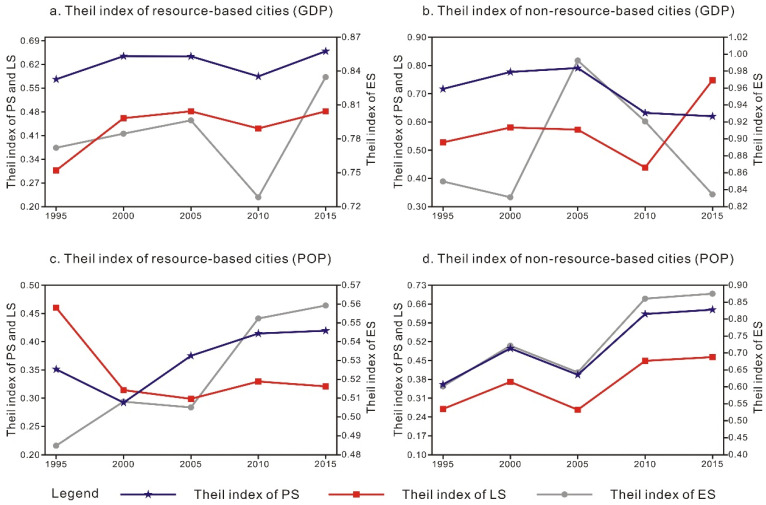
Change trend of the PLES-related Theil index of the two types of cities, 1995–2015. Note: The left ordinate indicates the PS and LS, and the right ordinate indicates the ES.

**Figure 6 ijerph-18-12497-f006:**
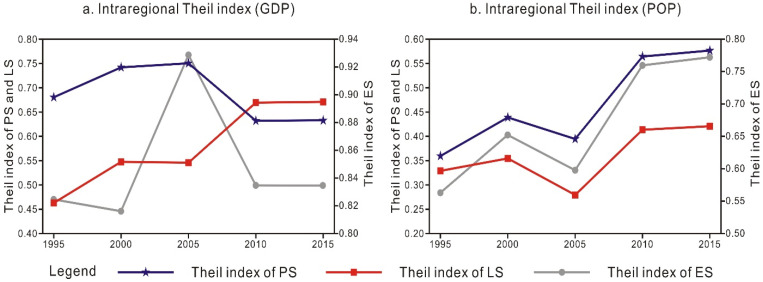
Change trend of the PLES-related Theil index in the region, 1995–2015. Note: The left ordinate indicates the PS and LS, and the right ordinate indicates the ES.

**Figure 7 ijerph-18-12497-f007:**
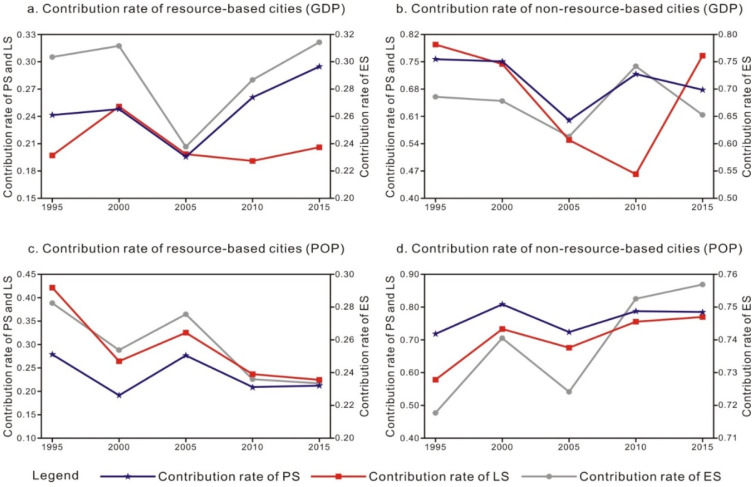
Change trend of the contribution rate of the PLES-related Theil index, 1995–2015. Note: The left ordinate indicates the PS and LS, and the right ordinate indicates the ES.

**Table 1 ijerph-18-12497-t001:** Data sources.

Type	Data Year	Data Source
Remote sensing data of land use/land cover in China (LUCC,1 km accuracy)	1995, 2000, 2005, 2010, 2015	Data center of resources and environment science, Chinese Academy of Sciences (http://www.resdc.cn/ accessed on 20 September 2021)
Population distribution of 1 km grid in China	1995, 2000, 2005, 2010, 2015	Data center of resources and environment science, Chinese Academy of Sciences (http://www.resdc.cn/ accessed on 20 September 2021)
GDP distribution of 1 km grid in China	1995, 2000, 2005, 2010, 2015	Data center of resources and environment science, Chinese Academy of Sciences (http://www.resdc.cn/ accessed on 20 September 2021)

**Table 2 ijerph-18-12497-t002:** Classification system of production-living-ecological space (PLES) of urban agglomeration in the middle reaches of the Yangtze River.

Primary Classification	Secondary Classification
Production space (PS)	Paddy field, dry land, other forest land (non-forest forest land, slash land, nursery and various gardens), other construction land (refer to land for factories and mines, large industrial areas, oil fields, salt fields, quarries, traffic roads, airports and special land)
Living space (LS)	Urban land (land for large, medium and small cities and built-up areas above counties and towns), rural residential areas (rural residential areas independent of cities and towns)
Ecological space (ES)	There are woodland (referring to natural forest and artificial forest with canopy density >30%), shrub forest (referring to dwarf forest and shrub forest with canopy density >40% and height below 2 m), sparse forest (referring to forest with canopy density of 10–30%), high coverage grassland (referring to natural grassland, improved grassland and mowed grassland with coverage >50%) and medium coverage grassland (refers to natural grassland and improved grassland with a coverage of 20–50%), low coverage grassland (refers to natural grassland with a coverage of 5–20%), rivers, lakes, reservoirs, ponds, permanent Glacial Snow, beaches, beaches, sandy lands, Gobi, saline alkali lands, swamps, bare lands, bare rocky lands, oceans and others

**Table 3 ijerph-18-12497-t003:** Change trend of the PLES of the two types of cities in the middle reaches of the Yangtze River, 1995–2015.

Year	Type	Moran’s I	Z Value	*p*-Value
1995				
	PS	0.397	11.681	0.001
	LS	0.438	13.031	0.000
	ES	0.492	14.404	0.000
2000				
	PS	0.39	11.459	0.000
	LS	0.441	13.131	0.000
	ES	0.491	14.376	0.000
2005				
	PS	0.39	11.459	0.000
	LS	0.423	12.461	0.002
	ES	0.49	14.348	0.000
2010				
	PS	0.316	9.318	0.000
	LS	0.386	11.472	0.002
	ES	0.43	12.242	0.000
2015				
	PS	0.378	11.113	0.000
	LS	0.432	12.775	0.000
	ES	0.49	14.322	0.000

**Table 4 ijerph-18-12497-t004:** Results for the PLES-related Theil index of the two types of cities, 1995–2015.

Weight Type	Year	Tr-PS	Tr-LS	Tr-ES	Tg-PS	Tg-LS	Tg-ES
GDP	1995	0.577	0.306	0.772	0.719	0.528	0.849
	2000	0.645	0.461	0.785	0.779	0.581	0.831
	2005	0.644	0.48	0.796	0.793	0.572	0.992
	2010	0.585	0.429	0.728	0.635	0.439	0.920
	2015	0.659	0.480	0.835	0.624	0.749	0.834
POP	1995	0.352	0.460	0.485	0.361	0.271	0.600
	2000	0.293	0.314	0.508	0.495	0.371	0.722
	2005	0.376	0.299	0.505	0.399	0.268	0.642
	2010	0.416	0.329	0.552	0.622	0.448	0.860
	2015	0.420	0.321	0.559	0.639	0.462	0.874

**Table 5 ijerph-18-12497-t005:** Interregional and intraregional Theil index values of the PLES, 1995–2015.

Weight Type	Year	Tw-PS	Tw-LS	Tw-ES	Tb-PS	Tb-LS	Tb-ES
GDP	1995	0.6785	0.4618	0.8241	0.0030	0.0050	0.0098
	2000	0.7411	0.5453	0.8157	0.0026	0.0042	0.0089
	2005	0.7504	0.5443	0.9281	0.1963	0.1847	0.1648
	2010	0.6311	0.6687	0.8342	0.0033	0.0018	0.0001
	2015	0.6314	0.6693	0.8342	0.0190	0.0204	0.0303
POP	1995	0.3585	0.3274	0.5624	0.0012	0.0004	0.0001
	2000	0.4378	0.3541	0.6519	0.0004	0.0012	0.0041
	2005	0.3922	0.2777	0.5973	0.0005	0.0001	0.0003
	2010	0.5641	0.4126	0.7587	0.0024	0.0041	0.0091
	2015	0.5753	0.4204	0.7714	0.0028	0.0033	0.0077

**Table 6 ijerph-18-12497-t006:** Contribution rate of the PLES-related Theil index of the two types of cities, 1995–2015.

Weight Type	Year	CTr-PS	CTr-LS	CTr-ES	CTg-PS	CTg-LS	CTg-ES
GDP	1995	0.245	0.197	0.303	0.755	0.793	0.685
	2000	0.247	0.250	0.311	0.749	0.742	0.678
	2005	0.196	0.198	0.238	0.597	0.548	0.612
	2010	0.261	0.191	0.286	0.718	0.460	0.741
	2015	0.295	0.206	0.314	0.676	0.765	0.651
POP	1995	0.278	0.421	0.282	0.718	0.578	0.718
	2000	0.191	0.264	0.253	0.808	0.733	0.740
	2005	0.275	0.324	0.276	0.724	0.675	0.724
	2010	0.208	0.235	0.236	0.788	0.755	0.752
	2015	0.212	0.224	0.233	0.784	0.768	0.757

## Data Availability

The data presented in this study are openly available in http://www.resdc.cn/.

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
