# Peer review of "Spatiotemporal Evolution and Regional Differences in the Production-Living-Ecological Space of the Urban Agglomeration in the Middle Reaches of the Yangtze River"

_ijerph, 2021, doi:10.3390/ijerph182312497_

Round 1

Reviewer 1 Report

The article raises the important issue of identification of the evolution and regiona differences in the production-living-ecological-space (PLES). The article is correctly written but contains several shortcomings that should be corrected.

General comments:

The introduction is written in a specialised way. Improve it, put it in a broader context and make it more comprehensible for a wider audience

The discussion of the results obtained is too short and does not refer to the results obtained by other researchers. Please put it in a broader context also in relation to the results obtained by other researchers.

Highlight what your study adds to the field is not clearly stated, this is very important

Specific comments:

Figure 1: Map without coordinates, please correct this, put the map in a broader context as readers from all over the world need to easily identify the research area where is located

Figure 2 Please improve the quality of the graphs and their scales

Figure 3 Please improve quality of maps, they are not clear. Give a detailed description so that they are standalone

Table 3. Is P'value still 0, if you give both to 3 decimal places, also give 0 to three places do it in a consistent way

Table 4 is sufficient to two significant places, I believe that in the other tables the values are sufficient to two significant places.

Table 3. Please correct the table so that the year is not in the type column

Line 391: a subchapter should not begin with a table

Line 396 this should be written in the description not separately

Line: 502 Remove that line

Author Response

Dear reviewer,

Thank you for considering our manuscript “Spatiotemporal evolution and regional differences in the production-living-ecological space of the urban agglomeration in the middle reaches of the Yangtze River”. We greatly appreciate the detailed and constructive reviews, as well as the opportunity to respond to reviewer comments.

We believe that all of the constructive reviewer comments have merited text amendments, which we outline in the point-by-point response below. All comments were addressed without significant changes to the scientific content and conclusions of the paper. Our revised manuscript has carefully and fully addressed the concerns of the reviewer, and we hope you agree that it is now suitable for publication in International Journal of Environmental Research and Public Health. I look forward to hearing from you in due course.

Yours sincerely,

Yanqiong Zhao

(On behalf of co-authors)

Response to Reviewer 1 Comments

Specific comments:

Point 1: Figure 1: Map without coordinates, please correct this, put the map in a broader context as readers from all over the world need to easily identify the research area where is located

Response 1: Thank for reviewer suggestions. We have redrawn Figure 1. to place the research area in a broader research context. Refer to line 152 for details.

Point 2: Figure 2 Please improve the quality of the graphs and their scales

Response 2: Thank for reviewer suggestions. We have adjusted the quality and scope of Figure 2. Refer to line 321 for details.

Point 3: Figure 3 Please improve quality of maps, they are not clear. Give a detailed description so that they are standalone.

Response 3: Thank for reviewer suggestions. We have improved the quality of Figure 3 and described it in detail. Refer to lines 355 to 360 for details.

Point 4: Table 3. Is P'value still 0, if you give both to 3 decimal places, also give 0 to three places do it in a consistent way

Response 4: Thank for reviewer suggestions. We have unified the P’value to three decimal places. Refer to line 340 for details.

Point 5: Table 4 is sufficient to two significant places. I believe that in the other tables the values are sufficient to two significant places.

Response 5: Thank for reviewer suggestions. Table 4. is results of PLES Theil index in two types of cities, 1995-2015. Table 5. is interregional and intraregional Theil index of PLES, 1995-2015. Table 6. is contribution rate of PLES Theil index in two types of cities, 1995-2015.

Point 6: Table 3. Please correct the table so that the year is not in the type column.

Response 6: The table number has been revised to Table 4. It has been corrected. Refer to line 387 for details. Many thanks.

Point 7: Line 391: a subchapter should not begin with a table.

Response 7: It has been replaced by reviewer’s suggestion. Refer to lines 412 to 417 for details. Many thanks.

Point 8: Line 396 this should be written in the description not separately.

Response 8: It has changed line in description. Refer to line 423 for details. Many thanks.

Point 9: Line: 502 Remove that line

Response 9: It has removed the line. Many thanks.

General comments:

The introduction is written in a specialised way. Improve it, put it in a broader context and make it more comprehensible for a wider audience

The discussion of the results obtained is too short and does not refer to the results obtained by other researchers. Please put it in a broader context also in relation to the results obtained by other researchers. Highlight what your study adds to the field is not clearly stated, this is very important.

Point 10: The introduction is written in a specialised way. Improve it, put it in a broader context and make it more comprehensible for a wider audience.

Response 10: Thank for reviewer suggestions. The introduction has been revised. The specific modification is in lines 120 to 130.

The researches on the spatial-temporal pattern and regional differences of the PLES of natural watersheds provide the basis for the land space control and natural watershed governance. Previous studies have important reference values for this paper, but most of them just studied the classification and temporal and spatial pattern of PLES. Furthermore, the research on the temporal and spatial pattern focused on the single scale of administrative division, such as national, provincial and urban scale. Few studies have analyzed the temporal and spatial pattern and regional differences of PLES in ecologically fragile natural watersheds. In this study, we aimed to reveal the temporal and spatial pattern and regional differences of the PLES of urban agglomerations in the middle reaches of the Yangtze River. Another aim was to make proposals for eco-environmental protection and high-quality development in the Yangtze River Basin.

Point 11: The discussion of the results obtained is too short and does not refer to the results obtained by other researchers. Please put it in a broader context also in relation to the results obtained by other researchers.

Response 11: Thank for reviewer suggestions. The discussion has been revised according to the comments of the reviewers. Refer to lines 467 to 492 for details.

In this study, we can conclude that in the urban agglomeration in the middle reaches of the Yangtze River, from 1995 to 2005, the synchronous increase of the PS and LS in resource-based cities squeezed the ES. From 2005 to 2010, the PS and LS of non-resource-based cities increased, squeezing the ES. From 2010 to 2015, the PS and LS of resource-based cities increased simultaneously, squeezing the ES. The LS and ES of the non-resource-based cities increased simultaneously, thus compressing the PS. The PLES of the two types of cities have the characteristics of spatial aggregation, and the density area is relatively stable. The regional differences in the PLES mainly origi-nate from intraregional differences. Economic development can better disturb the changes of PLES.

Cui et al., by analyzing the evolution characteristics of PLES pattern in Hubei Province, concluded that from 2009 to 2015, the PS and LS in Hubei Province expand-ed significantly, and the ES shrank significantly [38]. The expansion of PS is mainly concentrated in Wuhan urban circle, which is consistent with the conclusion that the PS of resource-based cities expanded and the ES was compressed from 2005 to 2015. Jin et al., by the research on the evolution and function measurement of PLES of urban agglomeration in Fujian Delta, concluded that the ES of urban agglomeration in Fujian delta showed the overall change characteristics of obvious reduction rather than expansion from 2000 to 2015 [33]. The main reason was the rapid development of social economy and the continuous expansion of urban land. This conclusion is consistent with the fact that economic development can more disturb the spatial change of PLES in this study. Yang et al., by studying the land use transformation and eco-environmental effects in the Yangtze River Delta, concluded that the production land decreased and the living land increased in the region from 1990 to 2010 [12]. In this study, the PS expansion in the middle reaches of the Yangtze River from 1990 to 2010. The reason may be that the development of agricultural production in the middle reaches of the Yangtze River is more developed than that in the Yangtze River Delta.

Point 12: Highlight what your study adds to the field is not clearly stated, this is very important.

Response 12: Thank for reviewer suggestions. We added this section. Refer to lines 127 to 130 for details.

Reviewer 2 Report

The article is well written, although in many parts it is assumed that the reader is as familiar with the subject as the authors. I recommend streamlining some parts in order to better describe the concepts used and why.

The results of the metric parameters developed in the methodology are not discussed and studied in the discussion. For example: "Appropriate policies should be formulated according to local conditions" which ones? 

The conclusions need to be rewritten and made more explicit by showing what the assumptions were and how they were validated or not in the text. It would be interesting to leave the study open-ended so that there can be debate on how to apply this study in other geographical contexts.

Place a map of geographical context: "Figure 1. Spatial distribution (...)"

In addition, figures and diagrams need a more accurate, concise and precise description.

Author Response

Dear reviewer,

Thank you for considering our manuscript “Spatiotemporal evolution and regional differences in the production-living-ecological space of the urban agglomeration in the middle reaches of the Yangtze River”. We greatly appreciate the detailed and constructive reviews, as well as the opportunity to respond to reviewer comments.

We believe that all of the constructive reviewer comments have merited text amendments, which we outline in the point-by-point response below. All comments were addressed without significant changes to the scientific content and conclusions of the paper. Our revised manuscript has carefully and fully addressed the concerns of the reviewer, and we hope you agree that it is now suitable for publication in International Journal of Environmental Research and Public Health. I look forward to hearing from you in due course.

Yours sincerely,

Yanqiong Zhao

(On behalf of co-authors)

The article is well written, although in many parts it is assumed that the reader is as familiar with the subject as the authors. I recommend streamlining some parts in order to better describe the concepts used and why. The results of the metric parameters developed in the methodology are not discussed and studied in the discussion. For example: "Appropriate policies should be formulated according to local conditions" which ones? The conclusions need to be rewritten and made more explicit by showing what the assumptions were and how they were validated or not in the text. It would be interesting to leave the study open-ended so that there can be debate on how to apply this study in other geographical contexts. Place a map of geographical context: "Figure 1. Spatial distribution (...)"In addition, figures and diagrams need a more accurate, concise and precise description.

Point 1: The results of the metric parameters developed in the methodology are not discussed and studied in the discussion. For example: "Appropriate policies should be formulated according to local conditions" which ones? 

Response 1: Thank for reviewer suggestions. We revised the discussion section. Meanwhile, we referred to the results obtained by other researchers and put it in a broader context also in relation to the results obtained by other researchers. We have revised “Appropriate policies should be formulated according to local conditions” into “Corresponding policies should be differentiated and formulated” and added the discussion of measurement parameter results. The specific description is in lines 467 to 515.

Point 2: The conclusions need to be rewritten and made more explicit by showing what the assumptions were and how they were validated or not in the text.

Response 2: Thank for reviewer suggestions. We rewrote the conclusion, with details on lines 517 to 553.

Via processing of spatiotemporal data related to the PLES of the resource- and non-resource-based cities in the urban agglomeration in the middle reaches of the Yangtze River from 1995 to 2015, this paper examines the spatial evolution and spatial aggregation characteristics of these two types of cities and dynamically assesses the regional differences in the PLES between these two types of cities under the influence of economic development and population distribution with the Theil index. The main conclusions are as follows:

(1) From 1995 to 2015, the PLES of resource-based cities and non-resource-based cities in the middle reaches of the Yangtze River changed sharply. In different periods of this process, the following phenomena exist in both types of cities. The ES of was compressed by the PS and LS, and the ES of the resource-based cities was compressed for a longer period. That was not conducive to ecological environment protection of the urban agglomeration in the middle reaches of the Yangtze River. The LS and ES mainly compressed the PS. And this phenomenon occurs for a longer time in non-resource-based cities, which did not facilitate economic production enhancement.

(2) From 1995 to 2015, the spatial aggregation characteristics of the PLES of the county-level cities in the urban agglomeration in the middle reaches of the Yangtze River were obvious, exhibiting positive spatial autocorrelation characteristics. From 1995 to 2015, the high-density areas (high-high category) and low-density areas (low-low category) of various PLES in resource-based cities and non-resource-based cities remained stable.

(3) The regional differences in the PLES of the urban agglomeration in the middle reaches of the Yangtze River mainly originated from the differences between urban areas. The PLES of the two types of cities in the urban agglomeration in the middle reaches of the Yangtze River was more sensitive to changes in economic development. The regional differences in the PLES of the non-resource-based cities are relatively large, while those in the PLES of the resource-based cities are relatively small. The possible main reason is that the economic development modes of the non-resource-based cities are diverse, while the development modes of the resource-based cities are relatively singular. The regional differences in the PS, LS and ES of the resource-based cities exert a major impact on the PLES of the urban agglomeration in the middle reaches of the Yangtze River.

In the study, Theil index and ESDA are used to explore the regional differences of PLES between the two types of cities under the influence of economic development and population distribution. However, the pattern, structure, form and relationship of the micro scale of PLES have not been deeply studied. In addition, the specific reasons leading to the change of PLES have not been deeply explored. For the above deficiencies, further exploration and research are needed.

Point 3: Place a map of geographical context: "Figure 1. Spatial distribution (...)"

Response 3: Thank for reviewer suggestions. Figure1. was redraw on line 152.

Point 4: In addition, figures and diagrams need a more accurate, concise and precise description.

Response 4: Thank for reviewer suggestions. We have added descriptions of figures and diagrams. The description of Figure 2. was described in lines 298 to 320. The description of Table 3. was described in lines 327 to 338. The description of Figure3,4. were described in lines 342 to 351 and lines 355 to 360. The description of Table 4. was described in lines 367 to 385. The description of Figure 5. was described in lines 392 to 410. The description of Table 5. was described in lines 412 to 418. The description of Figure 6. was described in lines 425 to 439. The description of Table 6. was described in lines 441 to 446. The description of Figure 7. was described in lines 452 to 465.

Reviewer 3 Report

Dear authors

I find your manuscript to be well-written, with a very compelling analysis. I would suggest expanding the discussion section, which is rather scanty in terms of arguments, with a view on production-living-ecological space research comparison with articles focused on other regions. I think that outlining the advantages of this method and shortcomings to be addressed in future research will increase the added value of your article to be used for cases studies in other regions. According to your reference list, the methodological proposal of production-living-ecological space is rather specific to China and could be uptaken by international research should its effectiveness be explained.

Author Response

Dear reviewer,

Thank you for considering our manuscript “Spatiotemporal evolution and regional differences in the production-living-ecological space of the urban agglomeration in the middle reaches of the Yangtze River”. We greatly appreciate the detailed and constructive reviews, as well as the opportunity to respond to reviewer comments.

We believe that all of the constructive reviewer comments have merited text amendments, which we outline in the point-by-point response below. All comments were addressed without significant changes to the scientific content and conclusions of the paper. Our revised manuscript has carefully and fully addressed the concerns of the reviewer, and we hope you agree that it is now suitable for publication in International Journal of Environmental Research and Public Health. I look forward to hearing from you in due course.

Yours sincerely,

Yanqiong Zhao

(On behalf of co-authors)

Response to Reviewer 3 Comments

Comments and Suggestions for Authors

I find your manuscript to be well-written, with a very compelling analysis. I would suggest expanding the discussion section, which is rather scanty in terms of arguments, with a view on production-living-ecological space research comparison with articles focused on other regions. I think that outlining the advantages of this method and shortcomings to be addressed in future research will increase the added value of your article to be used for cases studies in other regions. According to your reference list, the methodological proposal of production-living-ecological space is rather specific to China and could be uptaken by international research should its effectiveness be explained.

Point 1: I would suggest expanding the discussion section, which is rather scanty in terms of arguments, with a view on production-living-ecological space research comparison with articles focused on other regions.

Response 1: Thank for reviewer suggestions. According to your suggestion, we expanded the discussion. Refer to lines 467 to 492 for details. We have added the following paragraphs.

In this study, we can conclude that in the urban agglomeration in the middle reaches of the Yangtze River, from 1995 to 2005, the synchronous increase of the PS and LS in resource-based cities squeezed the ES. From 2005 to 2010, the PS and LS of non-resource-based cities increased, squeezing the ES. From 2010 to 2015, the PS and LS of resource-based cities increased simultaneously, squeezing the ES. The LS and ES of the non-resource-based cities increased simultaneously, thus compressing the PS. The PLES of the two types of cities have the characteristics of spatial aggregation, and the density area is relatively stable. The regional differences in the PLES mainly origi-nate from intraregional differences. Economic development can better disturb the changes of PLES.

Cui et al., by analyzing the evolution characteristics of PLES pattern in Hubei Province, concluded that from 2009 to 2015, the PS and LS in Hubei Province expand-ed significantly, and the ES shrank significantly [38]. The expansion of PS is mainly concentrated in Wuhan urban circle, which is consistent with the conclusion that the PS of resource-based cities expanded and the ES was compressed from 2005 to 2015. Jin et al., by the research on the evolution and function measurement of PLES of urban agglomeration in Fujian Delta, concluded that the ES of urban agglomeration in Fujian delta showed the overall change characteristics of obvious reduction rather than expansion from 2000 to 2015 [33]. The main reason was the rapid development of social economy and the continuous expansion of urban land. This conclusion is consistent with the fact that economic development can more disturb the spatial change of PLES in this study. Yang et al., by studying the land use transformation and eco-environmental effects in the Yangtze River Delta, concluded that the production land decreased and the living land increased in the region from 1990 to 2010 [12]. In this study, the PS expansion in the middle reaches of the Yangtze River from 1990 to 2010. The reason may be that the development of agricultural production in the middle reaches of the Yangtze River is more developed than that in the Yangtze River Delta.

Point 2: I think that outlining the advantages of this method and shortcomings to be addressed in future research will increase the added value of your article to be used for cases studies in other regions.

Response 2: Thank you for your advice. We added the advantages and disadvantages of the method in the conclusions part. Refer to lines 548 to 553 for details.

In the study, Theil index and ESDA are used to explore the regional differences of PLES between the two types of cities under the influence of economic development and population distribution. However, the pattern, structure, form and relationship of the micro scale of PLES have not been deeply studied. In addition, the specific reasons leading to the change of PLES have not been deeply explored. For the above deficiencies, further exploration and research are needed.

Reviewer 4 Report

The paper conducts an interesting research on spatiotemporal evolution and regional differences in the production-living-ecological space of the urban agglomeration in the middle reaches of the Yangtze River. However, there are still some aspects that can be refined. Authors are advised as follows.

Line 95: it would be interesting to understand in which direction has the PS changed in the Henan Province. Did it expand, contract or just change its spatial patterns?

Figure 1: it would be helpful for the reader to highlight some target reference points (i.e. the river, boundaries of Wuhan Urban Circle, Changsha Zhuzhou Xiangtan 123Urban Agglomeration and Poyang Lake Urban Agglomeration, etc.)

Table 3: Be aware there are 2 tables named "Table 3". In both cases, it would be better to introduce the table with a short paragraph explaining what it is going to illustrate.

Line 352: something is not working with the sentence "With the use of the Theil index and decomposition equations, this paper analyses the grid data capturing the county-level cities in the urban agglomeration in the middle reaches of the Yangtze River from 1995 to 2015, calculates the Theil index of the cities in the urban agglomeration in the middle reaches of the Yangtze River with the GDP and population as weights and then determines the regional differences between the above two types of cities".

Table 4: it would be better to introduce the table with a short paragraph explaining what it is going to illustrate.

Table 5: it would be better to introduce the table with a short paragraph explaining what it is going to illustrate.

Patents: what is this paragraph about? I see no content.

Author Response

Dear reviewer,

Thank you for considering our manuscript “Spatiotemporal evolution and regional differences in the production-living-ecological space of the urban agglomeration in the middle reaches of the Yangtze River”. We greatly appreciate the detailed and constructive reviews, as well as the opportunity to respond to reviewer comments.

We believe that all of the constructive reviewer comments have merited text amendments, which we outline in the point-by-point response below. All comments were addressed without significant changes to the scientific content and conclusions of the paper. Our revised manuscript has carefully and fully addressed the concerns of the reviewer, and we hope you agree that it is now suitable for publication in International Journal of Environmental Research and Public Health. I look forward to hearing from you in due course.

Yours sincerely,

Yanqiong Zhao

(On behalf of co-authors)

Comments and Suggestions for Authors

The paper conducts an interesting research on spatiotemporal evolution and regional differences in the production-living-ecological space of the urban agglomeration in the middle reaches of the Yangtze River. However, there are still some aspects that can be refined. Authors are advised as follows.

Point 1: Line 95: it would be interesting to understand in which direction has the PS changed in the Henan Province. Did it expand, contract or just change its spatial patterns?

Response 1: The PS of Henan Province has contracted on the whole from 1990 to 2015. The expansion area is mainly distributed in the central region of Henan Province, and the reduction area is mainly distributed in the urban areas of cities under the jurisdiction of each province. Among them, there is a large regional production space in the mountainous area in the south of Xinyang.

Point 2: Figure 1: it would be helpful for the reader to highlight some target reference points (i.e. the river, boundaries of Wuhan Urban Circle, Changsha Zhuzhou Xiangtan 123Urban Agglomeration and Poyang Lake Urban Agglomeration, etc.)

Response 2: It has been revised. Refer to line 152 for details. More thanks.

Point 3: Table 3: Be aware there are 2 tables named "Table 3". In both cases, it would be better to introduce the table with a short paragraph explaining what it is going to illustrate.

Response 3: It has been revised. An introduction to Table 3. has been added. Refer to lines 367 to 385 for details. More thanks.

Point 4: Line 352: something is not working with the sentence "With the use of the Theil index and decomposition equations, this paper analyses the grid data capturing the county-level cities in the urban agglomeration in the middle reaches of the Yangtze River from 1995 to 2015, calculates the Theil index of the cities in the urban agglomeration in the middle reaches of the Yangtze River with the GDP and population as weights and then determines the regional differences between the above two types of cities".

Response 4: This sentence has been corrected. Refer to lines 367 to 371 for details. More thanks.

“Using Theil index and its decomposition formula, we analyze the grid data of urban agglomeration in the middle reaches of the Yangtze River from 1995 to 2015, get the Theil index with GDP and population as the weight of the urban agglomeration in the middle reaches of the Yangtze River and calculate the regional differences between the two types of cities”.

Point 5: Table 4: it would be better to introduce the table with a short paragraph explaining what it is going to illustrate.

Response 5: Thank for reviewer suggestions. An introduction to Table 4. has been added. Refer to lines 367 to 385 for details.

Point 6: Table 5: it would be better to introduce the table with a short paragraph explaining what it is going to illustrate.

Response 6: Thank for reviewer suggestions. An introduction to Table 5. has been added. Refer to lines 412 to 418 for details.

Point 7: Patents: what is this paragraph about? I see no content.

Response 7: This part is wrong. It has been deleted.

Round 2

Reviewer 2 Report

The article has been revised and can be accepted as it stands.